# Ehrenfeucht-Haussler Rank and Chain of Thought

**Pablo Barceló** [* 1 2 3]   **Alexander Kozachinskiy** [* 3]   **Tomasz Steifer** [* 4]

## Abstract

The notion of *rank* of a Boolean function has been a cornerstone in PAC learning, enabling quasipolynomial-time learning algorithms for polynomial-size decision trees. We present a novel characterization of rank, grounded in the well-known Transformer architecture. We show that the rank of a function $f$ corresponds to the minimum number of *Chain of Thought* (CoT) steps required by a single-layer Transformer with hard attention to compute $f$. Based on this characterization we establish tight bounds on the number of CoT steps required for specific problems, showing that $\ell$-fold function composition necessitates exactly $\ell$ CoT steps. Furthermore, we analyze the problem of identifying the position of the $k$-th occurrence of 1 in a Boolean sequence, proving that it requires $k$ CoT steps.

## 1. Introduction

Ehrenfeucht & Haussler introduced the notion of the *rank* of a Boolean function and showed that, for any constant $r$, the class of Boolean functions with rank at most $r$ is properly PAC-learnable in polynomial time. As a corollary, they derived their renowned quasipolynomial-time PAC-learning algorithm for polynomial-size decision trees. Pudlák & Impagliazzo further characterized the rank—not only for Boolean functions but also for Boolean relations—through Prover-Delayer games. Since its introduction, this concept has played a significant role in proof complexity (Kullmann, 1999; Esteban & Torán, 2003).

In this paper, we present a new characterization of the notion of rank. Surprisingly, this characterization is grounded in the

---
[*]Equal contribution  [1]Institute for Mathematical and Computational Engineering, Pontifical Catholic University of Chile [2]Millennium Institute for Foundational Research on Data (IMFD Chile) [3]National Center for Artificial Intelligence (CENIA Chile) [4]Institute of Fundamental Technological Research, Polish Academy of Sciences. Correspondence to: Alexander Kozachinskiy <alexander.kozachinskyi@cenia.cl>.

*Proceedings of the 42ⁿᵈ International Conference on Machine Learning*, Vancouver, Canada. PMLR 267, 2025. Copyright 2025 by the author(s).

*Transformer architecture* (Vaswani et al., 2017), which has recently revolutionized the field of NLP and facilitated the development of LLMs. In essence, we show that the rank of a function $f$ corresponds to the minimum number of *Chain of Thought* (CoT) steps required by a single-layer Transformer to compute $f$. The Transformers used in our characterization are based on the *hard attention* mechanism—a theoretical abstraction of the *soft attention* mechanism employed in practice. Hard attention has been widely used in theoretical studies (Hahn, 2020b; Hao et al., 2022b; Barceló et al., 2024; Yang et al., 2024) due to its amenability to formal analysis, while still effectively capturing the essence of practical models (Clark et al., 2019; Voita et al., 2019).

The Transformer architecture is built upon *attention* layers and a *decoder*. An attention layer performs attention on the input sequence, mapping a sequence of input vectors to another sequence of vectors of the same length. Attention layers are used to generate vector representations of sentences in natural language. However, a more common application of Transformers is *sequence generation*, where the input sequence is mapped to an unbounded sequence of output vectors, generated iteratively, one at a time. This task is carried out by the decoder. In the first iteration, the decoder processes the input sequence through the attention layers and outputs the vector in the last position. This output is then appended to the input sequence. During subsequent iterations, the decoder applies its attention layers to the extended sequence, computes the next output, and appends it to the sequence. These are the CoT steps mentioned earlier (Merrill & Sabharwal, 2024; Liu et al., 2024).

Below we summarize our main results:

- We show that the rank of a function $f$, denoted by $\mathsf{rk}(f)$, is the minimal number of iterations of a single-layer decoder with one hard-attention head that computes $f$. We establish our result not only for Boolean functions, generalizing the notion of the rank to the non-Boolean case (as far as we know, for the first time).

- In practice, Transformers are equipped with multiple attention heads, which enhance their computational capabilities. We show that the ability of such Transformers to compute functions can also be characterized using the notion of rank. Specifically, we define the

*H-head rank* of a function $f$, denoted as $\mathsf{rk}^{(H)}(f)$, for $H \geq 1$. We prove that $\mathsf{rk}^{(H)}(f)$ equals the minimum number of iterations required by a single-layer decoder with $H$ hard-attention heads to compute $f$.

- We then explore methods for obtaining tight bounds on the multi-head rank. We begin by observing that $\mathsf{rk}^{(H)}(f)$ is at most a factor of $H$ smaller than $\mathsf{rk}(f)$. While computing $\mathsf{rk}(f)$ is typically straightforward, it does not always provide an accurate bound for $\mathsf{rk}^{(H)}(f)$. To address this limitation, we propose a general communication complexity lower bound for $\mathsf{rk}^{(H)}(f)$. Using this technique, we derive a tight bound on the $H$-head rank for the *t-fold iterated composition*, a function whose complexity has been previously studied for single-layer decoders with soft attention (Peng et al., 2024). The function $t$-Comp takes as input a sequence of $n$ integers from $\{1, \ldots, n\}$, interpreted as the values of a function $\phi \colon \{1, \ldots, n\} \to \{1, \ldots, n\}$. The output of $t$-Comp is the value of $\phi$, composed with itself $t$ times, evaluated at 1.

  It is easy to see that $\mathsf{rk}(t\text{-Comp}) \leq t$ for any input length $n$. A decoder, establishing this upper bound works by computing $\phi(1)$ in the first iteration, then $\phi(\phi(1))$ in the second iteration, and so on. We prove that this is optimal even if we increase the number of attention heads. Namely, for any $H$, we show that $\mathsf{rk}^{(H)}(t\text{-Comp}) = t$ for all large enough input lengths.

- Finally, we study the $k$-thOne function. This function takes as input a Boolean sequence of length $n$, and it returns the position of the $k$-th one in it. It is easy to see that $\mathsf{rk}(k\text{-thOne}) \leq k$ for any input length. In terms of decoders, in the first iteration we can compute the position of the first one, then of the second one in the second iteration, and so on. We prove that for any $H$ and for large enough $n$, we have $\mathsf{rk}^{(H)}(k\text{-thOne}) = k$, showing that even increasing the number of attention heads we cannot improve upon the trivial solution for large enough input lengths. Interestingly, this result cannot be obtained via the communication complexity techniques used for iterated composition. Instead, our proof relies on a purely combinatorial argument.

**Related work.** Numerous studies have sought to explore the expressive power of Transformers by treating them as a computational model and investigating what they can compute (Hahn, 2020a; Pérez et al., 2021; Hao et al., 2022a; Angluin et al., 2023; Chiang et al., 2023; Merrill & Sabharwal, 2023; Barceló et al., 2024; Merrill & Sabharwal, 2024; Liu et al., 2024; Yang & Chiang, 2024; Peng et al., 2024). In particular, several works have investigated how the capability of decoders depends on the number of iterations. To start with, Pérez et al. showed that decoders

based on hard attention with an unbounded number of iterations are capable of computing any decidable language (with the parameters of the decoder not depending on the input length). Afterwards, the computation power of decoders with polynomially many iterations was addressed. Merrill & Sabharwal have shown that in the uniform-regime (when, as in (Pérez et al., 2021), parameters do not depend on the input length), such decoders with constant number of layers and softmax attention are capable of computing any polynomial-time language. Similarly, for the non-uniform regime, (Liu et al., 2024) have shown that such decoders are capable of computing any language recognizable by a polynomial-size family of Boolean circuits.

Our result is the first *exact* characterization of the expressive power of decoders with a given fixed number of iterations, although just for a single layer and for hard attention. Recently, Peng et al. have shown that any single-layer decoder with soft attention requires $\Omega(t)$ iterations to compute $t$-Comp for $t = \sqrt{n/(dHp)}$, where $n$ is the input length, $d$ is the dimension of vectors, $H$ is the number of attention heads, and $p$ is the number of bits of precision. We point out that our results instead do not require any assumptions on the dimension and the number of bits of precision.

**Organization of the paper.** An introduction to decision trees and the notion of rank is found in Section 2, with basic concepts of Transformers being discussed in Section 3. The main results about single-head Transformers are presented in Section 4, with extensions to multi-head Transformers covered in Section 5. Final remarks are given in Section 6. Missing proofs can be found in the arXiv version of this paper.

## 2. Decision Trees and Rank

Consider $n + 1$ finite sets $\Sigma_1, \ldots, \Sigma_n, O$, for $n > 0$. We are interested in decision trees that compute functions:

$$f \colon \Sigma_1 \times \Sigma_2 \times \ldots \times \Sigma_n \to O.$$

To do this, we consider decision trees over arbitrary families of *queries*, where a query is a function $q$ whose domain is $\Sigma_1 \times \ldots \times \Sigma_n$. We write $\mathrm{Im}(q)$ for the image of query $Q$. If $\mathcal{F}$ is a set of queries, a decision tree over $\mathcal{F}$ is a rooted tree $T$ such that:

- Every non-leaf node $v$ is labeled by some query $q_v \in \mathcal{F}$ and has exactly $|\mathrm{Im}(q_v)|$ out-going edges, each one of them labeled by a different element from $\mathrm{Im}(q_v)$.

- Every leaf $\ell$ is labeled by some element $o_\ell \in O$.

Given an input $\bar{w} = (\sigma_1, \ldots, \sigma_n) \in \Sigma_1 \times \ldots \times \Sigma_n$, the output of decision tree $T$ on $\bar{w}$ is computed by descending

from the root to one of the leaves. At each intermediate non-leaf node $v$, the tree computes the value $q_v(\bar{w}) \in \text{Im}(q_v)$ and descends to the unique child of $v$ that is linked to $v$ through an edge labeled $q(\bar{w})$. In this way, we reach some leaf $\ell$, where $T$ outputs the element $o_\ell$ as its result on $\bar{w}$. We denote this output as $T(\bar{w})$.

The function $f : \Sigma_1 \times \ldots \times \Sigma_n \to O$ is *computed* by $T$, if $T(\bar{w}) = f(\bar{w})$ for every input $\bar{w} \in \Sigma_1 \times \ldots \times \Sigma_n$.

**Boolean case.** Decision trees are often defined for *Boolean* functions, i.e., functions of the form $f : \{0, 1\}^n \to \{0, 1\}$. In our notation, this corresponds to the case $\Sigma_1 = \ldots = \Sigma_n = O = \{0, 1\}$. *Boolean decision trees* are decision trees over a family $\{p_1, \ldots, p_n\}$ of queries, where for $i = 1, \ldots, n$ the function $p_i : \{0, 1\}^n \to \{0, 1\}$ is defined as follows on input $(b_1, \ldots, b_n) \in \{0, 1\}^n$:

$$p_i(b_1, \ldots, b_n) = b_i.$$

That is, at every node, a Boolean decision tree queries the value of some coordinate of the input.

Ehrenfeucht & Haussler defined the *rank* of a Boolean decision tree $T$ by inductively defining the rank of its nodes as follows:

- the rank of a leaf is $0$, and

- the rank of a non-leaf $v$, whose two children have ranks $r_0, r_1$, is $r = \max\{\min\{r_0, r_1\} + 1, \max\{r_0, r_1\}\}$.

The rank of $T$ is then the rank of its root, and the rank of a Boolean function $f : \{0, 1\}^n \to \{0, 1\}$ is the minimum rank of a Boolean decision tree that computes $f$.

**Rank in the non-boolean case and a-queries.** We extend the notion of rank to the non-Boolean case through decision trees over *assignment queries*. We start by introducing some terminology. Pairs of the form $(i, \sigma)$, where $i \in [n]$ and $\sigma \in \Sigma_i$, are called *assignments*. We denote by

$$A = \{1\} \times \Sigma_1 \cup \cdots \cup \{n\} \times \Sigma_n$$

the set of assignments. An assignment $(i, \sigma)$ is *consistent* with an input $\bar{w} = (\sigma_1, \ldots, \sigma_n) \in \Sigma_1 \times \ldots \Sigma_n$ if and only if $\sigma_i = \sigma$. By a permutation of a finite set $B$ we mean a bijection $\tau : \{1, \ldots, |B|\} \to B$.

An assignment query (*a-query* from now on) is a function of the form $q_\tau : \Sigma_1 \times \ldots \times \Sigma_n \to A$, where $\tau$ is a permutation of the set of assignments $A$. For $\bar{w} \in \Sigma_1 \times \ldots \times \Sigma_n$, we let $k_{\bar{w}}$ be the minimal element $k \in \{1, \ldots, |A|\}$ such that $\tau(k)$ is consistent with $\bar{w}$. We then define $q_\tau(\bar{w}) = \tau(k_{\bar{w}})$.

It is sometimes convenient to view the computation of an a-query $q_\tau$ on an input $\bar{w}$ as follows. Assume that

$\tau(j) = (i_j, \sigma_j)$, for each $j = 1, \ldots, |A|$. Imagine that we do not know $\bar{w}$, and we start asking a person who knows $\bar{w}$ questions: "is the $i_1$-th letter of $\bar{w}$ equal to $\sigma_1$?", 'is the $i_2$-th letter of $\bar{w}$ equal to $\sigma_2$?", and so on. We stop once we receive the first YES answer. If this happens at the $k$th step, we return $q_\tau(\bar{w}) = (i_k, \sigma_k)$.

We define the rank of an arbitrary function $f : \Sigma_1 \times \ldots \times \Sigma_n \to O$ in terms of the class of decision trees over assignment queries that compute $f$.

**Definition 2.1.** Let $f : \Sigma_1 \times \ldots \times \Sigma_n \to O$. We define $\text{rk}(f)$ as the minimal depth of a decision tree over a-queries that computes $f$. □

As we show below, the notion of rank we have just introduced for arbitrary functions aligns, in the case of Boolean functions, with the definition we previously provided for that class of functions.

**Proposition 2.2.** *For any Boolean function $f : \{0, 1\}^n \to \{0, 1\}$, its rank, as defined by Ehrenfeucht and Haussler, is equal to $\text{rk}(f)$.*

**An example: Iterated composition** We consider the *iterated composition function*. We use a notation $[n] = \{1, \ldots, n\}$ for $n \in \mathbb{N}$. For positive integer numbers $t, n$, we define:

$$t\text{-Comp}_n : [n]^n \to [n],$$
$$t\text{-Comp}_n : (f(1), \ldots, f(n)) \mapsto \underbrace{f(f(\ldots f(1)))}_{t \text{ times}}.$$

A clarification for the second line: an input to $t\text{-Comp}_n$ is an $n$-length word, where every letter is a number from 1 to $n$. This input is interpreted as a function $f : [n] \to [n]$, with $f(1)$ being the first letter of the word, $f(2)$ being the second letter of the word, and so on. Sometimes, we also use the following notation:

$$f^{(\ell)} = \underbrace{f \circ f \circ \ldots \circ f}_{\ell \text{ times}}.$$

In particular, we let $f^{(0)}$ be the identity function.

We claim that the rank of $t\text{-Comp}_n$ does not exceed $t$. Recall that the input is interpreted as a word $(f(1), \ldots, f(n))$, for some $f : [n] \to [n]$, and our task is to compute $f^{(t)}(1)$. Consider a decision tree that first tries to guess the value of the first letter, that is, of $f(1)$ by going "is $f(1) = 1$?", "is $f(1) = 2$?", and so on. Once the tree gets it right, receiving the first YES-answer, it already knows $f(1)$, and now it starts guessing the $f(1)$st letter, that is, $f^{(2)}(1) = f(f(1))$. It costs the second YES-answer to get it right. Continuing in this way, the tree will find out $f^{(t)}(1)$ after $t$ YES-answers.

By means of a combinatorial argument, it is possible to show that this is the best one can do if $n$ is large enough.

**Proposition 2.3.** *For any $t$ and for all $n > 2t$, we have* $\mathsf{rk}(t\text{-}\mathsf{Comp}_n) = t$.

*Proof.* Assume for contradiction that we have a decision tree $T$ of depth $t-1$ over a-queries for $t\text{-}\mathsf{Comp}_n$, for some $n > 2t$. We start answering questions for $T$, descending to one of its leafs, in the following manner. We maintain a set $F \subseteq [n]$ of "forbidden numbers". Initially, $F = \{1\}$. When we receive an a-query with a permutation $\tau$ of assignments, we select the first assignment $(i, j)$ such that $j \notin F$ and $f(i)$ is not fixed yet. We fix $f(i) = j$ and continue along the tree as if this was the first consistent assignment. After that, we put $j$ into $F$. Note that after $k$ values of $f$ have been fixed this way, $F$ consists of precisely $k + 1$ distinct elements. Indeed, every a-query we consider adds exactly one new element to $F$.

Let $\ell$ denote the leaf of $T$ where we come in this way by answering a-queries. Suppose that $o_\ell \in [n]$ is the value that $T$ outputs in this leaf. We obtain a contradiction by showing that some function $g \colon [n] \to [n]$ with $g^{(t)}(1) \neq o_\ell$ also gets to $\ell$.

Observe that, since $T$ is of depth $t-1$, there are $k \leq t-1$ a-queries on the path to $\ell$ and the same number of values of $f$ have been fixed:

$$f(i_1) = j_1, \ f(i_2) = j_2, \ldots, f(i_k) = j_k. \qquad (1)$$

Note that $i_1, \ldots, i_k$ are distinct because we never fix the same value twice. Numbers $j_1, \ldots, j_k$ are distinct too, and they define the evolution of the set $F$. Initially, $F = \{1\}$ after the first a-query, $F = \{1, j_1\}$ after the second a-query, $F = \{1, j_1, j_2\}$ after the third one, and so on.

Take any $y \in [n] \setminus \{1, i_1, \ldots, i_k, j_1, \ldots, j_k, o_\ell\}$ (it exists because $n > 2t \geq 2(k+1)$). Define a function $g \colon [n] \to [n]$ by

$$g(i_1) = j_1, \ldots, g(i_k) = j_k,$$
$$g(x) = y \text{ for } x \in [n] \setminus \{i_1, \ldots, i_k\}.$$

We first show that $g$ arrives to $\ell$ in $T$. For that, we show that $g$ is consistent with all answers to questions on the path to $\ell$. All the assignments corresponding to our answers to a-queries on the path to $\ell$ are as in (1), and $g$ is consistent with all of them by definition. Next, take an assignment $(i, j)$ and suppose it appears at the $m$-th a-query along the path to $\ell$, ordered before the assignment $(i_m, j_m)$ (which we chose to be the first consistent one). Hence, in our descent along the tree we ignored this assignment and decided to fix the assignment $(i_m, j_m)$ instead. Hence, we need to observe that $g$ is not consistent with it, that is, that $g(i) \neq j$. Indeed, we could have ignored $(i, j)$ in two cases. Firstly, it could have happened that $g(i)$ was already fixed to some value different to $j$. Secondly, we could have ignored it when $g(i)$

was not yet fixed, because $j$ already belonged to the set of forbidden numbers $F$. But by definition of $g$ that means that either $g(i) = y$ or $g(i) = j_s$ for some $s > m$. The first case is not possible since $y$ was chosen to be outside of $F$, and the second case gives us $g(i) \neq j$.

To finish the proof, we show that $g^{(t)}(1) = y$. Consider a directed graph with vertex set $\{1, \ldots, n\}$, where for every $i \in \{1, \ldots, n\}$ there is a directed edge from $i$ to $g(i)$. The image of the function $g$ consists of $j_1, \ldots, j_k$ and $y$. In the graph, these are the only nodes with incoming edges. Observe that each of $j_1, \ldots, j_k$ has exactly one incoming edge. Namely, for $s = 1, \ldots, k$, the node $j_s$ has a unique incoming edge from $i_s$. To compute $g^{(t)}(1)$, we start moving from 1 along the edges for $t$ steps. We will be moving over $j_1, \ldots, j_k$ and $y$. Note that $g(y) = y$ because $y \notin \{i_1, \ldots, i_k\}$. Hence, it is enough to show that $y$ is reached from 1 in *at most* $t$ steps because then we stay at $y$ forever. Now, if we do not reach $y$ within the first $t$ steps, then we travel over $j_1, \ldots, j_k$ for $t$ steps. Since $k \leq t-1$, it means that we come into some of $j_1, \ldots, j_k$ two times, but this would mean that one of them has two distinct incoming edges, which is impossible. $\square$

**An example: Position of the $k$-th one.** We define a function $k\text{-}\mathsf{thOne}_n \colon \{0, 1\}^n \to [n+1]$ such that:

$$k\text{-}\mathsf{thOne}_n(\sigma_1, \ldots, \sigma_n) =$$
$$\min\left(\{n+1\} \cup \{i \in [n] : \sigma_1 + \ldots + \sigma_i = k\}\right).$$

In other words, given $\bar{w} = (\sigma_1, \ldots, \sigma_n) \in \{0, 1\}^n$, the function $k\text{-}\mathsf{thOne}_n$ returns the position of the $k$-th one in $\bar{w}$ (counting from the left). If there are fewer than $k$ ones in $\bar{w}$, we return $n + 1$. We can then show the following by means of a combinatorial argument:

**Proposition 2.4.** *For any $n, k$, we have* $\mathsf{rk}(k\text{-}\mathsf{thOne}_n) \leq k$, *and for $n \geq k^2 + k$, we have* $\mathsf{rk}(k\text{-}\mathsf{thOne}_n) = k$.

## 3. Attention Layers and Decoders

**Attention layer.** We consider layers with *unique hard attention*, and possibly multiple attention heads, where the output of the layer is computed in the last token. By unique hard attention we refer to the mechanism in which each position attends to the element with the highest attention score (breaking ties arbitrarily).

Formally, a *unique hard-attention layer* (or, simply, attention layer) with $H$ heads and embedding dimension $d$ is a function $L \colon (\mathbb{R}^d)^* \to \mathbb{R}^d$, which is defined by

- $H$ *query* matrices $Q^{(h)} \in \mathbb{R}^{d \times d}$ and $H$ *key* matrices $K^{(h)} \in \mathbb{R}^{d \times d}$, for $h = 1, \ldots, H$,

- two matrices $W_1, W_2 \in \mathbb{R}^{d \times d}$, and

- a matrix $W_O \in \mathbb{R}^{d \times (dH)}$.

Consider an input sequence of vectors $(x_1, \ldots, x_m) \in (\mathbb{R}^d)^m$. The output of $L$ on $(x_1, \ldots, x_m)$ is computed as follows. For every $h = 1, \ldots, H$, we compute the *value of the h-th head* on $(x_1, \ldots, x_m)$, which is a vector from $\mathbb{R}^d$ denoted by $\mathrm{head}_h \in \mathbb{R}^d$. Namely, we start by computing "attention scores"

$$a_{i,m}^{(h)} = \langle K^{(h)} x_i, Q^{(h)} x_m \rangle, \tag{2}$$

defining, for every $i = 1, \ldots, m$, the *attention* from the last token to the $i$-th token with respect to the $h$-th head. The vector $K^{(h)} x_i$ is called the *key* of the $i$-th token, and the vector $Q^{(h)} x_m$ is called the *query* of the $m$-th token.

For every $h = 1, \ldots, H$, we let $i_h \in \{1, \ldots, m\}$ to be the index maximizing (2). If there are multiple indices achieving the maximum, we let $i_h$ be the leftmost one. We then set $\mathrm{head}_h = x_{i_h}$, for $h = 1, \ldots, H$, and define:

$$\mathrm{multihead} = W_O \cdot \begin{pmatrix} \mathrm{head}_1 \\ \vdots \\ \mathrm{head}_H \end{pmatrix} \in \mathbb{R}^d \tag{3}$$

Finally, we define:

$$
\begin{aligned}
&L(x_1, \ldots, x_m) \\
&\quad = W_2 \cdot \mathrm{ReLU}\left(W_1(\mathrm{multihead} + x_m)\right) \in \mathbb{R}^d. \tag{4}
\end{aligned}
$$

Recall that $\mathrm{ReLU}(x) = \max\{0, x\}$, for every $x \in \mathbb{R}$, and if $x \in \mathbb{R}^d$ then $\mathrm{ReLU}(x)$ is obtained by applying ReLU to each one of its components.

**Decoders.** A *decoder*, defined by the $d$-dimensional attention layer $L$, is a function that takes on input a sequence of vectors $(x_1, \ldots, x_m) \in (\mathbb{R}^d)^m$ and in the output produces an infinite sequence of vectors $\{y_t \in \mathbb{R}^d\}_{t=1}^{\infty}$, defined by:

$$
\begin{aligned}
y_1 &= L(x_1, \ldots, x_m), \\
y_t &= L(x_1, \ldots, x_m, y_1, \ldots, y_{t-1}), \qquad t \geq 2.
\end{aligned}
$$

That is, the decoder works in iterations: first, it computes the output of $L$, adds it to the end of the input sequence, computes the output of $L$ on the new sequence, adds this output to the end, and so on. We refer to $y_t$ as the output of the decoder after $t$ iterations (sometimes these iterations are called "chain of thought steps").

**Computation of functions by decoders.** Fix $n$ and $n+1$ finite sets $\Sigma_1, \ldots, \Sigma_n, O$. We want to define how a decoder computes functions of the form:

$$f : \Sigma_1 \times \ldots \times \Sigma_n \to O.$$

Inputs to $f$ are interpreted as words with $n$ letters, with the $i$-th letter coming from the alphabet $\Sigma_i$, for $i = 1, \ldots, n$ (alphabets are possibly different at different positions). We put this word as an input to a decoder using $n+1$ tokens, one per letter plus a special token at the end for the "end of line" symbol. Input tokens can use arbitrary encodings of letters by $d$-dimensional vectors, potentially different at different positions of the input word, utilizing in this form a *positional* information. We then run the decoder on the resulting input for some number $t$ of iterations. The output of $f$ is computed by applying an output function to the decoder's output $y_t$ from the final iteration.

**Definition 3.1** (Computation of functions by decoders). Let $n$ be a natural number and $\Sigma_1, \ldots, \Sigma_n, O$ be $n+1$ finite sets. A function $f : \Sigma_1 \times \ldots \times \Sigma_n \to O$ can be *computed by $t$ iterations of a decoder with $H$ heads*, if there exist:

- $d \in \mathbb{N}$ and an attention layer $L$ of embedding dimension $d$ with $H$ heads,

- a *positional encoding* $p$, i.e. a function $p : \Sigma_1 \times \{1\} \cup \ldots \cup \Sigma_n \times \{n\} \cup \{\mathsf{EoL}\} \to \mathbb{R}^d$, where $\mathsf{EoL}$ denotes a special "end-of-line" symbol, and

- an *output function* $\alpha : \mathbb{R}^d \to O$,

such that for any $\bar{w} = (\sigma_1, \ldots, \sigma_n) \in \Sigma_1 \times \ldots \times \Sigma_n$, the value $f(\bar{w})$ is determined by the following procedure:

1. Define a sequence $(x_1, \ldots, x_n, y_0)$ of $d$-dimensional vectors by:

   $$x_1 = p(\sigma_1, 1), \ \ldots, x_n = p(\sigma_n, n), \ y_0 = p(\mathsf{EoL}).$$

2. Place $(x_1, \ldots, x_n, y_0)$ as an input to the the decoder defined by $L$, and let $y_t$ for $t \geq 1$ denote the output of this decoder after $t$ iterations.

3. Set $f(\bar{w}) = \alpha(y_t)$. $\qquad\square$

Next, we define the following important notion.

**Definition 3.2** (Decoder depth of a function). The *decoder depth with $H$ heads* of $f : \Sigma_1 \times \ldots \times \Sigma_n \to O$, denoted $\mathsf{dd}^{(H)}(f)$, is the minimum $t \geq 0$ such that $f$ can be computed by $t$ iterations of a decoder with $H$ heads. $\qquad\square$

## 4. One-Head Decoder Depth vs Tree Rank

In this section, we show that the rank of a function is equivalent to its decoder depth in the single-head setting.

**Theorem 4.1.** *For any function $f : \Sigma_1 \times \ldots \Sigma_n \to O$, we have $\mathsf{rk}(f) = \mathsf{dd}^{(1)}(f)$.*

As a corollary to Theorem 4.1 and Proposition 2.3, we obtain that for suitable $n$ the decoder depth with one head of the iterated composition function $t\text{-}\mathsf{Comp}_n$ is precisely $t$:

**Corollary 4.2.** *For each $t$ and for all $n > 2t$, we have* $\mathsf{dd}^{(1)}(t\text{-}\mathsf{Comp}_n) = t$.

Also, as a corollary to Theorem 4.1 and Proposition 2.4, we obtain that for suitable $n$ the decoder depth with one head of the $k$th one function $k\text{-}\mathsf{thOne}_n$ is precisely $k$:

**Corollary 4.3.** *For each $k$, and for every $n \geq k^2 + k$, we have* $\mathsf{dd}^{(1)}(k\text{-}\mathsf{thOne}_n) = k$.

We now prove our main theorem.

*Proof of Theorem 4.1.* We first show the inequality $\mathsf{rk}(f) \leq \mathsf{dd}^{(1)}(f)$. Assume that $f$ can be computed by a decoder with one head in $r$ iterations, for some $r \in \mathbb{N}$. We deduce that $\mathsf{rk}(f) \leq r$. For that, we show that at the cost of $t$ a-queries one can compute the outputs of the decoder in the first $t$ iterations on a given input. Hence, in $r$ a-queries, we can compute the $r$th output of the decoder, which uniquely determines the value of $f$, implying that $\mathsf{rk}(f) \leq r$.

Consider any input $\bar{w} = (\sigma_1, \ldots, \sigma_n) \in \Sigma_1 \times \ldots \times \Sigma_n$. Define then:

$$x_1 = p(1, \sigma_1), \ldots, x_n = p(n, \sigma_n), \ y_0 = p(\mathsf{EoL}) \in \mathbb{R}^d,$$

where $d$ is the dimension of our decoder and $p$ is its positional encoding function. Let $\{y_t \in \mathbb{R}^d\}_{t=1}^{\infty}$ be the sequence of the outputs of our decoder on input $(x_1, \ldots, x_n, y_0)$. Assume that we have already computed $y_1, \ldots, y_t$ for some $t \geq 0$ (if $t = 0$, we just know $y_0 = p(\mathsf{EoL})$). We explain how to compute $y_{t+1}$ using one a-query. By definition,

$$y_{t+1} = L(x_1, \ldots, x_n, y_0, y_1, \ldots, y_t),$$

where $L$ is the attention layer defining our decoder. It is enough to compute $s \in \{x_1, \ldots, x_n, y_0, y_1, \ldots, y_t\}$ with the maximal value of $\langle Ks, Qy_t \rangle$ for the key and query matrices $K, Q \in \mathbb{R}^{d \times d}$ of our attention layer. If there are multiple vectors $s \in \{x_1, \ldots, x_n, y_0, \ldots, y_t\}$ with the maximal value of this scalar product, we need to compute the leftmost one among them. Since we already have computed $y_0, y_1, \ldots, y_t$, it suffices to find this maximal $s$ over $\{x_1, \ldots, x_n\} = \{p(1, \sigma_1), \ldots, p(n, \sigma_n)\}$.

Consider the following linear order of the set $A$ of assignments. Given two different assignments $a = (i, \sigma)$, $a' = (i', \sigma')$, we say that $a$ is larger than $a'$ if either $\langle Kp(a), Qy_t \rangle > \langle Kp(a'), Qy_t \rangle$ or $\langle Kp(a), Qy_t \rangle = \langle Kp(a'), Qy_t \rangle$ and $i < i'$. We arbitrarily order assignments with $\langle Kp(a), Qy_t \rangle = \langle Kp(a'), Qy_t \rangle$ and $i = i'$. Our task is to find the maximal assignment from $\{p(1, \sigma_1), \ldots, p(n, \sigma_n)\}$ in this order. For that, we ask the a-query $q_\tau$ for a permutation $\tau$, where the first assignment is the maximal in our linear order, the second one is the second maximal, and so on.

We now show the inequality $\mathsf{dd}^{(1)}(f) \leq \mathsf{rk}(f)$. Assume that $T$ is an $r$-depth decision tree over a-queries that computes $f$. We transform into a decoder with one head that computes $f$ in $r$ iterations. We assume that $T$ is a complete $r$-depth $|A|$-ary tree, where $A$ is the set of assignments.

The embedding dimension of our decoder will be:

$$\begin{aligned} d = &\ 1 + |A| + \ldots + |A|^{r-1} \\ &+ 1 + |A| + \ldots + |A|^r \\ &+ |A| \\ &+ 1. \end{aligned}$$

The coordinates will be split into 4 groups:

- the first $1 + |A| + \ldots + |A|^{r-1}$ coordinates are called *positional coordinates* and are indexed by non-leaf nodes of $T$;

- the second $1 + |A| + \ldots + |A|^r$ coordinates are called *output coordinates* and are indexed by nodes of $T$;

- the third $|A|$ coordinates are called *assignment coordinates* and are indexed by assignments;

- the last coordinate will be called *special*.

Our goal is to construct a decoder that "simulates" $T$ in the following sense. On input $\bar{w} \in \Sigma_1 \times \ldots \times \Sigma_n$, for any $t \geq 0$, we want the $t$-th output of the decoder, denoted by $y_t \in \mathbb{R}^d$, to be the one-hot encoding of the node where $T$ comes on $\bar{w}$ at depth $t$. More specifically, this one-hot encoding will take place in output coordinates, the remaining coordinates of $y_t$ will all be 0.

To achieve this, we start with defining $y_0 = p(\mathsf{EoL}) \in \mathbb{R}^d$ as follows. In the restriction to the output coordinates it is the one-hot encoding of the root of $T$; all the other coordinates of $y_0$ are 0. Next, we define the positional encoding $p(a) \in \mathbb{R}^d$ for an assignment $a = (i, \sigma) \in A$. In the restriction to the assignment coordinates, it is the one-hot encoding of $a$. Now, for each non-leaf node $v$ of $T$ and its corresponding positional coordinate $p(a)_v$, we set $p(a)_v = 1/\tau_v^{-1}(a)$, where $\tau_v \colon \{1, \ldots, |A|\} \to A$ is the permutation defining the a-query asked at $v$. We let the special coordinate of $p(a)$ to be 1. Finally, all output coordinates of $p(a)$ are set to 0.

Having our positional encoding defined, we move to the construction of the attention layer and define the query matrix $Q \in \mathbb{R}^{d \times d}$ by the following linear transformation $\alpha \mapsto Q\alpha$ for $\alpha \in \mathbb{R}^d$: for every non-leaf node $v$ of $T$, the $v$-th positional coordinate of $Q\alpha$ is equal the $v$-th output coordinate of $\alpha$; the remaining coordinates of $Q\alpha$ are 0. The key matrix $K \in \mathbb{R}^{d \times d}$ is set to be the identity matrix.

Assume that, as an input, for some $t < r$, we give to a layer the following sequence of vectors:

$$x_1, \ldots, x_n, \; y_0, y_1, \ldots, y_t \in \mathbb{R}^d,$$

where $x_i = p(i, \sigma_i)$ for $i = 1, \ldots, n$ and for some $\bar{w} = (\sigma_1, \ldots, \sigma_n) \in \Sigma_1 \times \ldots \times \Sigma_n$, $y_0 = p(\mathsf{EoL})$, and for every $i = 1, \ldots, t$, the vector $y_i$ is the one-hot encoding, inside the output coordinates, of some depth-$i$ node $v_i$ of $T$, and has 0 in the remaining coordinates. Let $q = q_{v_t}$ be the a-query asked at $v_t$, and let $\tau = \tau_{v_t}$ be the corresponding permutation of the set of assignments (the node $v_t$ is a non-leaf node because $t < r$). We claim that the attention on this input will be maximized for the position with the assignment which is the output of $q$ on $\bar{w}$.

Indeed, the vector $y_t$ has the unique 1 at the $v_t$-th output coordinate, with the remaining coordinates of $v_t$ being 0. The matrix $Q$ moves this 1 into the $v_t$-th positional coordinate, and the rest of the coordinates of $Qy_t$ are 0. Thus, for any $\alpha \in \mathbb{R}^d$, the product $\langle K\alpha, Qy_t \rangle$ equals the value of $\alpha$ in the $v_t$-th positional coordinate. If $\alpha = p(i, \sigma_i)$ for $i \in [n]$, this value is $1/\tau^{-1}(i, \sigma_i)$. The maximum of this value is attained for $(i, \sigma_i) \in \{(1, \sigma_1), \ldots, (n, \sigma_n)\}$ with the minimal value of $\tau^{-1}(i, \sigma_i)$, i.e. for $(i, \sigma_i) = q(\bar{w})$. Now, for $\alpha \in \{y_0, y_1, \ldots, y_t\}$, the value of the $v_t$-th positional coordinate, as well as any other positional coordinate, is 0. Hence, the output of the head will be the vector $p(q(\bar{w}))$.

The output of the layer is now computed as:

$$y_{t+1} = W_2 \cdot \mathrm{ReLU}\left(W_1 \cdot \beta\right), \tag{5}$$
$$\beta = p(q(\bar{w})) + y_t. \tag{6}$$

We need to choose $W_1, W_2 \in \mathbb{R}^{d \times d}$ such that the resulting $y_{t+1}$ will encode the node $v_{t+1}$ where the tree goes from $v_t$ by following the $q(\bar{w})$-labeled edge. More specifically, we want $y_{t+1}$ to be the one-hot encoding of $v_{t+1}$ in the output coordinates, and we want all the other coordinates of $y_{t+1}$ to be 0. We will set $W_2$ to be the identity matrix. To define $W_1$, we fix the following notation. For a non-root node $v$ of $T$, let $parent(v)$ denote the parent node of $v$, and let $label(v) \in A$ denote the label of the edge from $parent(v)$ to $v$. We define $W_1$ by the following linear transformation $\alpha \mapsto W_1\alpha, \alpha \in \mathbb{R}^d$: for every non-root node $v$ of $T$, we define the $v$-th output coordinate of $W\alpha$ as

$$\text{the } parent(v)\text{-th output coordinate of } \alpha \tag{7}$$
$$+ \;\; \text{the } label(v)\text{-th assignment coordinate of } \alpha \tag{8}$$
$$- \;\; \text{the special coordinate of } \alpha. \tag{9}$$

We set all the other coordinates of $W_1\alpha$ to 0.

We have to show now that $\mathrm{ReLU}(W_1 \cdot \beta)$, with $\beta$ as in (5–6) has 1 in the $v_{t+1}$-th output coordinate and 0 in all the other coordinates. Indeed, $W_1 \cdot \beta$ has 0 in any coordinate which

is not an output coordinate for a non-root node of $T$. Now, consider any non-root node $v$ of $T$. It is enough to show that the $v$-th output coordinate of $W_1 \cdot \beta$ is 1 for $v = v_t$ and is 0 or -1 for $v \neq v_t$ (applying ReLU to 0 and $-1$, we get 0).

To calculate the $v$-th output coordinate of $W_1\beta$, as stated in (7–9), we calculate the $parent(v)$-th output coordinate of $\beta$, the $label(v)$-th assignment coordinate of $\beta$, and the special coordinate of $\beta$. Recall that positional encodings of assignments have 0 in the output coordinates. Hence, the sum $\beta = p(q(\bar{w})) + y_t$, in the restriction to the output coordinates, is the one-hot encoding of $v_t$. In other words, the $parent(v)$-th output coordinate of $\beta$ is the indicator $\mathbb{I}\{parent(v) = v_t\}$. Likewise, since $y_t$ has only 0 in the non-output coordinates, the sum $\beta = p(q(\bar{w})) + y_t$, in the restriction to the assignment coordinates, is the one-hot encoding of the assignment $q(\bar{w})$. Again, this means that the $label(v)$-th assignment coordinate of $\beta$ is equal to the indicator $\mathbb{I}\{label(v) = q(\bar{w})\}$. Finally, the special coordinates of $p(q(\bar{w}))$ and $y_t$ are 1 and 0, respectively, meaning that the special coordinate of $\beta$ is 1. Plugging these equalities into (7–9) for $\alpha = \beta$, we obtain that the $v$-th output coordinate of $W_1\beta$ equals:

$$\mathbb{I}\{parent(v) = v_t\} + \mathbb{I}\{label(v) = q(\bar{w})\} - 1.$$

This expression takes values in $\{-1, 0, 1\}$ and it is equal to 1 if and only if both indicators are 1. It remains to note that $v_{t+1}$ is the only node whose parent is $v_t$ and such that the label of the edge from $v_t$ to this node is $q(\bar{w})$.

The $r$-th output of the decoder, $y_r$, in restriction to the output coordinates, will be the one-hot encoding of the leaf to which we come while computing $T$ on input $\bar{w}$. Since this leaf uniquely determines $f(\bar{w})$, we are done. $\qquad\square$

## 5. Multihead Rank

In order to generalize Theorem 4.1 to decoders with many heads, we define the notion of $H$-head rank for a function $f : \Sigma_1 \times \ldots \times \Sigma_n \to O$. For that we require a notion of the *product* of two functions with the same domain. Namely, by the product of $f \colon A \to B$ and $g \colon A \to C$, we mean a function $(f \otimes g) \colon A \to B \times C$, defined by:

$$(f \otimes g)(a) = (f(a), g(a)).$$

An *$H$-degree a-query* is a product of $H$ a-queries.

**Definition 5.1.** The $H$-head rank of a function $f : \Sigma_1 \times \ldots \times \Sigma_n \to O$, denoted $\mathsf{rk}^{(H)}(f)$, is the minimal depth of a decision tree over $H$-degree a-queries that computes $f$.

A simple generalization of the construction of Theorem 4.1 allows us to obtain the following result.

**Theorem 5.2.** *For any $H \in \mathbb{N}$ and for any function $f : \Sigma_1 \times \ldots \times \Sigma_n \to O$, we have $\mathsf{rk}^{(H)}(f) = \mathsf{dd}^{(H)}(f)$.*

We observe that the $H$-head rank can be at most $H$ times smaller than the normal rank. Specifically, each $H$-degree a-query can be computed by performing $H$ individual a-queries sequentially.

**Proposition 5.3.** *For $f : \Sigma_1 \times \ldots \times \Sigma_n \to O$ and $H \geq 1$, we have* $\mathsf{rk}(f) \leq H \cdot \mathsf{rk}^{(H)}(f)$.

Proposition 5.3 allows us to reduce, up to a factor of $H$, lower bounds on $\mathsf{rk}^{(H)}(f)$ to lower bounds on $\mathsf{rk}(f)$. However, this proposition is sometimes unable to provide tight bounds on $\mathsf{rk}^{(H)}(f)$. This occurs, for instance, when $\mathsf{rk}^{(H)}(f)$ is not smaller at all than $\mathsf{rk}(f)$. We present two examples of this phenomenon in this section.

To establish precise lower bound on the decoder depth of a function with $H$ heads, it suffices to derive a lower bound on its $H$-head rank (Theorem 5.2). However, this task proves to be significantly more challenging than for the single-head rank. Specifically, for the iterated composition function, combinatorial arguments alone, as employed in the proof of Proposition 2.3, are no longer sufficient. Instead, we must rely on techniques from communication complexity to address the problem. For the $k\text{-thOne}_n$ function, we develop a combinatorial argument that is notably more intricate than the one used in the proof of Proposition 2.4.

### 5.1. Multihead decoder depth of iterated composition

In this section, we show a method for lower bounding the multihead rank of a function based on communication complexity (Kushilevitz & Nisan, 1996). Let $\mathcal{X}, \mathcal{Y}, \mathcal{Z}$ be finite sets and $f : \mathcal{X} \times \mathcal{Y} \to \mathcal{Z}$ be a function. Imagine that there are two players, Alice and Bob. Alice is given $x \in \mathcal{X}$ and Bob is given $y \in \mathcal{Y}$. Their goal is to cooperatively compute $f(x, y)$. For that, they can send each other messages that are binary words. They want to minimize the number of messages and their total length in bits.

Formally, a *$k$-round Alice-first communication protocol* $\Pi$ is given by:

- $k$ positive integer numbers $\ell_1, \ldots, \ell_k$ (messages lengths);

- a function $M_i : \{0,1\}^{\ell_1 + \ldots + \ell_{i-1}} \times \mathcal{X} \to \{0,1\}^{\ell_i}$ for every odd $i \in \{1, \ldots, k\}$;

- a function $M_i : \{0,1\}^{\ell_1 + \ldots + \ell_{i-1}} \times \mathcal{Y} \to \{0,1\}^{\ell_i}$ for every even $i \in \{1, \ldots, k\}$; and

- the output function $out : \{0,1\}^{\ell_1 + \ldots + \ell_k} \to \mathcal{Z}$.

The *communication complexity* of $\Pi$ is the sum $\ell_1 + \ldots + \ell_k$.

On input $(x, y) \in \mathcal{X} \times \mathcal{Y}$, the *output* of $\Pi$ on $(x, y)$ is computed as follows. We inductively define a sequence of

binary words $m_1 \in \{0,1\}^{\ell_1}, \ldots, m_k \in \{0,1\}^{\ell_k}$ by setting

$$m_i = M_i(m_1 \ldots m_{i-1}, x) \text{ for odd } i \in \{1, \ldots, k\},$$
$$m_i = M_i(m_1 \ldots m_{i-1}, y) \text{ for even } i \in \{1, \ldots, k\}.$$

Intuitively, $m_1 = M_1(\varepsilon, x)$ is the first message of Alice that she sends to Bob in the protocol on input $x$. Upon receiving $m_1$, Bob replies with the second message $m_2 = M_2(m_1, y)$ that depends on his input and the first of Alice's messages. Then Alice sends the third message $m_3 = M_3(m_1 m_2, x)$, and so on. The output of the protocol is defined as $out(m_1 \ldots m_k) \in \mathcal{Z}$.

By $C^{k,A}(f)$ we mean the minimal communication complexity of a *$k$-round Alice-first protocol* that computes $f$. By reversing the roles of Alice and Bob, we define *$k$-round Bob-first protocols*, and $C^{k,B}(f)$, the minimal communication complexity of a $k$-round Bob-first protocol for a function $f$.

Assume we have a function $f : \Sigma_1 \times \ldots \times \Sigma_n \to O$ and a subset $S \subseteq [n]$. Suppose that positions of an input word $\bar{w} \in \Sigma_1 \times \ldots \times \Sigma_n$ are split between Alice and Bob like this: Alice knows letters of $\bar{w}$ at positions $i \in S$, and Bob knows letter of $\bar{w}$ at positions $i \in [n] \setminus S$. Their goal is to find out $f(\bar{w})$. This defines a function:

$$f^S : \left( \prod_{i \in S} \Sigma_i \right) \times \left( \prod_{i \in [n] \setminus S} \Sigma_i \right) \to O,$$

where the two inputs correspond to the parts of $\bar{w}$ that Alice and Bob knows, respectively, and the output of is $f(\bar{w})$.

Assuming that the $H$-head rank of $f$ is $r$, we construct low-communication $(r + 1)$-round Alice-first and Bob-first protocols for $f^S$, for any $S \subseteq [n]$. This gives a method for lower bounding the multihead rank of $f$: by showing that either $C^{r+1,A}(f)$ and $C^{r+1,B}$ is large enough, we conclude that the $H$-head rank of $f$ is larger than $r$.

**Lemma 5.4.** *For every $f : \Sigma_1 \times \ldots \times \Sigma_n \to \{0,1\}$, for every $S \subseteq [n]$, and for every $H \geq 1$, denoting $r = \mathsf{rk}^{(H)}(f)$ and $|A|$ the number of assignments for $f$, we have:*

$$C^{r+1,A}(f^S) \leq 2Hr \cdot \lceil \log_2 |A| \rceil \quad and$$
$$C^{r+1,B}(f^S) \leq 2Hr \cdot \lceil \log_2 |A| \rceil.$$

*Proof.* We first notice that Alice and Bob can compute the value of any $H$-degree a-query $q_{\tau_1} \otimes \ldots \otimes q_{\tau_H}$ by exchanging messages of length $H \cdot \lceil \log_2 a \rceil$. In fact, for a given input $\bar{w} \in \Sigma_1 \times \ldots \times \Sigma_n$ there are exactly $n$ assignments consistent with $\bar{w}$. A part of them is known to Alice (for positions in $S$) and the other part to Bob (for positions in $[n] \setminus S$). For each $h = 1, \ldots, H$, Alice and Bob have to calculate the first assignment in the permutation $\tau_h$ which is consistent with $\bar{w}$. Alice can see which $\bar{w}$-consistent assignment, known to her, goes first in $\tau_h$, and the same for Bob.

Among these two assignments, the one that goes first is the answer to $q_{\tau_h}$. Alice and Bob just have to exchange the indices of these assignments. For both Alice and Bob it is thus enough to send a $H\lceil \log_2 a \rceil$-bit message with indices of $H$ assignments.

We already see that an $r$-depth decision tree over $H$-degree a-queries can be simulated by a communication protocol with $2Hr \cdot \lceil \log_2 a \rceil$ bits. We need to explain how to arrange this communication in $r + 1$ rounds. For that, Alice and Bob have to alternate the order in which they exchange their messages in a computation of the $H$-degree a-queries. For example, for the Alice-first protocol, in the computation of the first query Alice has to send her message first and then Bob. Now, for the second query, *Bob has to send his message first* and then Alice. In this way, Bob's messages for the first and for the second query merge into a single round of communication. Similarly, for the third query, Alice has to send first, and then Bob, and so on, getting overall $r + 1$ rounds. The Bob-first protocol is constructed in an analogous fashion. $\square$

As a corollary, we obtain the following:

**Corollary 5.5.** *For every $H$ and $t$, for all but finitely many $n$, we have* $\mathrm{rk}^{(H)}(t\text{-}\mathsf{Comp}_n) = t$.

*Proof.* We reduce from a communication problem called *pointer chasing*. In this problem, Alice is given $f_A \colon \{1, \ldots, m\} \rightarrow \{1, \ldots, m\}$ and Bob is given $f_B \colon \{1, \ldots, m\} \rightarrow \{1, \ldots, m\}$. In the $k$-step pointer chase, denoted here by $\mathrm{PC}_k^m$, the goal of Alice and Bob is to compute:

$$\underbrace{\ldots f_A(f_B(f_A(1))\ldots)}_{k \text{ times}}$$

It is easy to see that $C^{k,A}(\mathrm{PC}_k^m) = O(k \log m)$ (Alice starts by sending $m_1 = f_A(1)$, Bob replies by sending $m_2 = f_B(m_1)$, and so on). It is known that this task requires much longer communication for $k$-round *Bob-first* protocols. Namely, for any constant $k$, we have $C^{k,B}(\mathrm{PC}_k) = \Omega(m)$ (Duris et al., 1987).

It remains to notice that $\mathrm{PC}_t^{n/2}$ is a special case of the problem $t\text{-}\mathsf{Comp}_n^S$, for $S = \{1, \ldots, n/2\}$, where Alice gets $(\phi(1), \ldots, \phi(n/2))$ and Bob gets $(\phi(n/2 + 1), \ldots, \phi(n))$, for some function $\phi \colon \{1, \ldots, n\} \rightarrow \{1, \ldots, n\}$, and the task is to compute $\phi^{(k)}(1)$. Namely, we obtain $\mathrm{PC}_t^{n/2}$ as a special case when $\phi$ maps the first half of the inputs into the second half, and the second half into the first half. Assuming that $\mathrm{rk}^{(H)}(t\text{-}\mathsf{Comp}_n) < t$, by Lemma 5.4 we obtain:

$$Omega(n) = C^{t,B}(\mathrm{PC}_t^{n/2}) \leq$$
$$C^{t,B}(t\text{-}\mathsf{Comp}_n^S) \leq 2Ht \cdot \lceil \log_2 n^2 \rceil.$$

For any fixed $H, t$ this is true only for finitely many $n$. $\square$

### 5.2. Multihead decoder depth of $k$th One

In this section, we establish a tight lower bound on the multi-head rank of $k$-thOne.

**Theorem 5.6.** *For any $k, H \in \mathbb{N}$, for all but finitely many $n \in \mathbb{N}$, we have* $\mathrm{rk}^{(H)}(k\text{-}\mathsf{thOne}_n) = k$.

We observe that our communication complexity tool is not applicable in this case, as for any partition of the input positions between Alice and Bob, there exists a 2-round protocol with logarithmic communication that computes the position of the $k$-th one: Alice sends the positions of the first $k$ ones in her part of the input, and Bob does the same.

**Proposition 5.7.** *For any $k, n$ and $S \subseteq [n]$:*

$$C^{2,A}(k\text{-}\mathsf{thOne}_n^S) = C^{2,B}(k\text{-}\mathsf{thOne}_n^S) = O(k \log n).$$

If we wanted to use Lemma 5.4 to obtain a lower on $\mathrm{rk}^{(H)}(k\text{-}\mathsf{thOne}_n)$, we would have needed $C^{2,A}(k\text{-}\mathsf{thOne}_n^S)$ or $C^{2,B}(k\text{-}\mathsf{thOne}_n^S)$ to grow super-logarithmically with $n$ for some $S \subseteq [n]$. Instead, we use a self-reducibility technique by means of partial fixations.

## 6. Final Remarks

We have shown that the expressive power of single-layer Transformers with hard attention is tightly connected to the notion of rank of functions. Extending this characterization to more layers or to soft attention is a challenging future direction. In a contemporaneous manuscript, Chen et al. have proved unconditional lower bounds on the embedding dimension of multilayer decoder-only Transformers with soft attention that compute iterated function composition. However, their version of the problem differs significantly from the one considered here: they have several functions to compose, and each function is completely given in a single token. We plan to explore whether the techniques used in their work can be applied to strengthen our results.

## Acknowledgements

Kozachinskiy is supported by ANID Fondecyt Iniciación grant 11250060. Barceló and Kozachinskiy are funded by the National Center for Artificial Intelligence CENIA FB210017, Basal ANID. Barceló is also funded by ANID Millennium Science Initiative Program Code ICN17002.

## Impact Statement

This paper presents work whose goal is to advance the field of Machine Learning. There are many potential societal consequences of our work, none of which we feel must be specifically highlighted here.

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
