# OpenReview forum: "Ehrenfeucht-Haussler Rank and Chain of Thought"
_ICML.cc/2025/Conference — ICML 2025 poster_

### Official Review · Reviewer_R7ST · 2025-03-13

**Overall Recommendation:** 3

**Summary:**

The paper studies the expressivity of one-layer Transformers with hard attention (in particular for Boolean functions), where it's shown that the Ehrenfeucht-Haussler rank of a Boolean function is equal to the number of (continuous) chain-of-thought (CoT) tokens that a single-head model has to produce to express the target label. The paper further extends this finding to multi-head single-layer Transformers and more general alphabets by defining appropriate ranks.

## update after rebuttal

I maintain my positive assessment for the paper.

**Claims And Evidence:**

The claims are supported by theoretical arguments and proofs. However, I have some concerns about the mathematical modeling of decoder-only Transformers in this paper. In particular, this paper assumes that CoT tokens are continuous arbitrary vectors and not predefined tokens. So I wonder if the results hold if we somehow limit the Transformer to a constant size (e.g., {0,1, EoL}) alphabet.

**Essential References Not Discussed:**

I think the original chain of thought and scratchpad papers should be cited.

**Experimental Designs Or Analyses:**

No experiments.

**Methods And Evaluation Criteria:**

See above.

**Other Comments Or Suggestions:**

- When discussing the related work, some results depend on the size and depth of the models. It would be nice to discuss them in more detail.
- Elaborating on the questions below could be helpful.

**Other Strengths And Weaknesses:**

The paper uses different notions and techniques from theoretical computer science. This makes the paper interesting but also hard to read. Nevertheless the writing is quite clear.

**Questions For Authors:**

- What is the central message of the paper for the ML community?
- Can the arguments work with a decoder model that predicts discrete tokens belonging to a constant-size alphabet? (See 'Claims and Evidence' part)
- I think the decision trees could be defined and explained better. Do trees always have exactly $n$ queries? Is it always the same query at a particular depth? What is the meaning of the equation on line 121?
- In line 209 what is $u_{i_h}$?

**Relation To Broader Scientific Literature:**

This paper studies the relation between CoT complexity and rank of Boolean functions for a specific class of Transformer functions. The paper does not provide any broad insight into the working mechanism of Transformers or large language models and is pretty shallow on the machine learning side. So I think the paper would be mainly of interest of a niche community of ML theory people.

**Theoretical Claims:**

Yes, the theoretical arguments seem valid to me. See above for remarks on modeling.

---

> ### Author Rebuttal · Authors · 2025-03-30
>
> Thank you a lot for your comments and questions. We respond to all three below.
>
> *Q1: What is the central message of the paper for the ML community?*
>
> The ability of Transformers to perform function composition has garnered increasing attention in recent years, as understanding this capability sheds light on the computational resources they require to infer implicit knowledge from a given set of facts. Peng et al. demonstrated that single-layer, soft-attention Transformers without Chain-of-Thought (CoT) reasoning are fundamentally incapable of function composition. However, when CoT is introduced, they can achieve iterated composition—albeit at the cost of requiring a growing number of steps, which depends on both vector dimensionality and feature precision. Our work precisely quantifies the number of steps needed for t-th iterated composition and establishes that, under the idealized assumption of hard-attention, the number of required CoT steps is exactly t. This finding underscores a key insight: while CoT enables function composition, it does so incrementally—one step at a time. We believe this to be the central message for the ML community.
>
> *Q2: Can the arguments work with a decoder model that predicts discrete tokens belonging to a constant-size alphabet? (See 'Claims and Evidence' part)*
>
> Our argument works with discrete chain-of-thought tokens as well. Indeed, in our construction, CoT tokens essentially are used to maintain the one-hot encoding of the current node of the decision tree. In place of one-hot encoding, we can store these nodes directly in discrete tokens.
> We do not know how to obtain our results with discrete tokens, belonging to the constant-size alphabet.  Note that in our setting, even input tokens do not necessarily belong to a constant-size alphabet (for example, in t-comp, input tokens come from the alphabet {1, …, n}).
>
> *Q3: In line 209 what is $u_{i_h}$?*
>
> This should be $x_{i_h}$. Thank you for noticing this typo.

---

> > ### Comment · Reviewer_R7ST · 2025-04-07
> >
> > Thank you for the response. In agreement with other reviewers, I think it would be beneficial to explain the relation of this work with other complexity measures. Also, I think the alphabet should be discussed more thoroughly (for example, if we use discrete tokens, how many of them would be needed?)

---

### Official Review · Reviewer_MAzv · 2025-03-14

**Overall Recommendation:** 4

**Summary:**

The paper shows that the minimal number of CoT steps a single-layer hard-attention Transformer needs to compute a function exactly corresponds to the function's EH rank. In essence, it proves that the EH rank equals the minimum depth of decision trees (over assignment queries) that simulate the Transformer's computation process. Two directions are established: one where Transformer iterations mimic decision tree steps to resolve attention choices, and another where a Transformer decoder is constructed to replicate a decision tree of a given rank.

Key canonical functions are analyzed. For iterated composition, it’s shown that the EH rank equals the number of compositions, while for the kth-one identification problem, the rank exactly equals k when the input size is sufficiently large. These results are supported by communication complexity and combinatorial arguments.

Additionally, the paper extends these findings to multi-head attention by defining an H-head rank, proving that even with parallel a-queries, the sequential steps required for tasks like composition or counting remain unchanged. Overall, the work bridges classical complexity (EH rank) with modern Transformer architectures, highlighting both their computational power and the limitations of parallelism in structured tasks.

**Claims And Evidence:**

The claims are supported by clear theoretical evidence within the paper’s scope (single-layer hard-attention Transformers). The proofs for equivalence and lower bounds are systematic, leveraging combinatorial fixation, communication complexity, and inductive self-reducibility. However, the reliance on hard attention and idealized input fixation limits practical applicability. No evident gaps or errors are present in the core arguments.

1. Equivalence of EH Rank and Decoder Depth: $\mathrm{rk}(f) = \mathrm{dd}^{(1)}(f)$ for any function $f$.
       1. Rank ≤ Decoder Depth: Simulates Transformer steps via decision trees over a-queries. Each CoT step corresponds to resolving one a-query (Appx. A.1).
       2. Decoder Depth ≤ Rank: Builds a Transformer decoder emulating rank-$r$ decision trees (Sec. 4). Positional encodings track decision paths, ensuring equivalence. The bidirectional reduction is explicit, with detailed embeddings and attention mechanisms (Sec. 4.1).

2. Tight Bounds for Canonical Functions $\rightarrow$ for t-Comp $\mathrm{rk}(t\text{-Comp}_n) = t$ for $n > 2t$.
Authors show combinatorial argument via input fixation (Prop. 2.3). Fixing $t-1$ values forces at least $t$ a-queries (Appx. A.2).

3. Authors also show Multi-Head Generalization claiming $\mathrm{rk}^{(H)}(f) = \mathrm{dd}^{(H)}(f)$, for $t$-Comp, reduces to pointer chasing (PC) with $\Omega(t)$ rounds (Cor. 5.5) and for $k$-thOne, uses induction and unfixed intervals (Thm. 5.6).

---
Few limitations that I found:

1. Results apply to idealized hard attention, not softmax-based models. The paper acknowledges this but does not explore extensions (Sec. 6).
2. Arguments for $k$-thOne (Appx. A.3) rely on intricate partial fixations. While valid, the self-reducibility step assumes $f(n)$ grows sufficiently without explicit bounds.
3. The reduction from PC assumes Bob-first protocols require $\Omega(n)$ communication, but the paper does not re-prove this result, relying on Duris et al. (1987).

**Essential References Not Discussed:**

The paper does not cite several recent works that provide critical context for its contributions. Below are key omissions:
## 1.
1.1 Li et al. (ICLR 2024): ["Chain of Thought Empowers Transformers to Solve Inherently Serial Problems"](https://openreview.net/forum?id=3EWTEy9MTM) proves that transformers with CoT can simulate polynomial-size circuits, resolving whether CoT fundamentally increases expressivity. This work establishes that CoT allows transformers to overcome parallel computation limits (e.g., processing **TC⁰** vs **AC⁰**), which directly contextualizes the current paper’s focus on sequential steps and rank equivalence.
1.2 The current paper’s claim that CoT steps correspond to EH rank aligns with Li et al.’s conclusion but lacks a broader complexity-theoretic framing.

## 2.
Feng et al. (2023): ["Transformers as Neural Solomoff Inducers"](https://arxiv.org/abs/2310.10691)* connects CoT to circuit complexity, showing transformers with CoT can solve problems outside NC (parallelizable classes).  The current paper’s combinatorial proofs for k-thOne and t-Comp could be strengthened by contrasting with Feng et al.’s circuit-depth arguments.


## 3.
Hu et al. (2024): ["Stepwise Self-Consistent Training for Language Agents"](https://arxiv.org/abs/2402.03286) demonstrates empirically that CoT steps reduce estimation error in multi-step reasoning, even with noisy intermediate tokens. The current paper’s theoretical analysis of hard attention could benefit from engaging with Hu et al.’s findings on robustness.

## 4.
Chen et al. (2024): ["Theoretical Limitations of Multi-Layer Transformer"](https://arxiv.org/abs/2412.02975) proves dimensional lower bounds for soft attention in deep transformers. While cited briefly in Sec. 6, this work is critical for understanding why the current paper’s single-layer, hard-attention analysis cannot trivially extend to multi-layer models.  The paper does not contrast its rank-based bounds with Chen et al.’s dimension-dependent constraints.

## 5.
Bhattamishra et al. (2024): ["Transformers Learn Higher-Order Programs"](https://arxiv.org/abs/2403.00732) shows transformers with CoT can learn program induction in-context.  The current paper’s focus on decision trees could be enriched by discussing how program induction aligns with EH rank’s sequential resolution.

**Experimental Designs Or Analyses:**

The paper is purely theoretical, focusing on proving equivalences and lower bounds via combinatorial/communication complexity arguments. Since it contains no empirical experiments, there are no traditional "experimental designs" or statistical analyses to critique.

1. All results assume idealized hard attention (argmax over tokens). While acknowledged, this limits practical relevance to real-world softmax-based Transformers.
2. Lower bounds rely on adversarial input constructions (worst-case analysis). No consideration of average-case or probabilistic inputs.
3. In Thm 4.1, matrices $W_1, W_2$ are described via logic (e.g., “ReLU correctly prunes invalid paths”) but lack explicit parameterization. While the logic holds, explicit matrices would strengthen rigor.
4. The $t\text{-Comp}$ lower bound (Cor 5.5) depends on Duris et al.’s $ \Omega(n) $ bound for PC. While this is a standard reference, the paper does not validate if $t\text{-Comp}$’s structure fully aligns with PC’s assumptions (e.g., cyclic dependencies).

**Methods And Evaluation Criteria:**

1. The equivalence between EH rank and decoder depth is established via explicit bidirectional reductions (decision trees ↔ Transformers). This provides a formal foundation for analyzing CoT steps.  Constructive embeddings (Sec. 4) ensure equivalence, while combinatorial fixation (Prop. 2.3) and communication complexity (Cor. 5.5) enforce tight bounds.

2. Reduction to pointer chasing (PC) with Ω(t) communication rounds (Duris et al., 1987) is valid and aligns with established complexity theory.Partial fixation arguments (Appx. A.3) enforce sequential resolution of 1s, leveraging inductive self-reducibility

Limitations:
1. The analysis is restricted to single-layer decoders. Modern Transformers use multiple layers, which may reduce CoT steps via hierarchical processing—unaddressed in the paper.
2. The decoder depth equivalence assumes custom positional encodings (Sec. 4.1). It is unclear if results hold for standard embeddings (e.g., sinusoidal, learned).
3. While the paper proves multi-head rank equivalences (Thm. 5.2), it does not identify functions where additional heads reduce steps. The claim that “multi-head attention cannot circumvent inherent sequential steps” applies only to t-Comp/k-thOne—not general tasks.
4. One thing which I was hoping to find in this paper: its applicability to real-world Transformers remains unproven. Future work should address soft attention and multi-layer models while incorporating standard NLP benchmarks.

**Other Comments Or Suggestions:**

No other questions or comments for the authors.

**Other Strengths And Weaknesses:**

The paper is a theoretically rigorous contribution that advances our understanding of Transformers’ computational limits. While its scope is narrow, its originality and formal insights lay groundwork for future research on CoT and neural architectures.

**Questions For Authors:**

**Question 1: Hard Attention vs. Practical Softmax Models**
The paper establishes equivalences under hard attention, but real-world Transformers use softmax attention. **How do the authors anticipate their results generalizing to soft attention?**

---

**Question 2: Combinatorial Fixation Assumptions**
The lower bound for $k$-thOne relies on adversarial input fixation (Lemma A.2). **Do these bounds hold for inputs with probabilistic structure (e.g., i.i.d. 1s) or only worst-case inputs?**

---

**Question 3: Multi-Layer Architectures**
The paper focuses on single-layer decoders. **Can the EH rank framework extend to multi-layer architectures, given Chen et al.’s (2024) dimensional lower bounds for multi-layer soft-attention models?**

---

**Question 4: Explicit Matrix Definitions**
Theorem 4.1’s Transformer construction lacks explicit parameterization of $W_1, W_2$. **Can the authors provide a full specification of these matrices (e.g., via block-diagonal structures or sparse encodings)?**

---

**Question 5: Relation to CoT’s Circuit-Theoretic Power**
The paper does not cite Li et al. (ICLR 2024), which shows CoT enables Transformers to solve **P**-complete problems. **How does EH rank align with CoT’s role in overcoming parallel computation limits (e.g., TC⁰ vs. AC⁰ separations)?**

**Relation To Broader Scientific Literature:**

The paper advances three major threads:

1. Bridging Complexity Theory and Transformers: Connects EH rank (PAC learning) to CoT steps (Transformer theory).
2. Multi-Head Limitations: Shows parallelism cannot circumvent sequential rank bounds, contrasting with softmax-based analyses.
3. Combinatorial Proof Techniques: Introduces novel fixation/self-reducibility arguments for positional tasks, complementing communication complexity methods.

These results refine the understanding of Transformer’s inherent limitations, providing theoretical grounding for empirical observations about CoT’s necessity in complex reasoning.

**Theoretical Claims:**

All the theoretical claims looks to be correct, just one doubt on Theorem 5.2 -> Multi-Head Rank Equivalence claims $ \mathrm{rk}^{(H)}(f) = \mathrm{dd}^{(H)}(f) $, each head’s a-query is parallelized via $ H $-degree queries. Positional encodings and matrices generalize the single-head case. $W_O$ concatenates head outputs, and $ W_1, W_2 $ ensure state transitions. Assumes ReLU correctly prunes invalid paths. This is valid if matrix dimensions align with the expanded coordinate system?

---

> ### Author Rebuttal · Authors · 2025-03-30
>
> Thank you kindly for all your comments and inspiring questions!
>
> *The claim that “multi-head attention cannot circumvent inherent sequential steps” applies only to t-Comp/k-thOne—not general tasks.*
>
> A following function with H-head rank 1 and 1-head rank H establishes tightness of proposition 5.3. [Dahiya, Mahajan, On (Simple) Decision Tree Rank] gives the function OR_H \comp AND_m, defined as the disjunction of H conjunction, where each conjunction is taken on m disjoint variables. The aforementioned paper shows that its normal rank is H. On the other hand, its H-head rank is 1 because each head can compute one conjunction.
>
> *All the theoretical claims looks to be correct, just one doubt on Theorem 5.2 -> Multi-Head Rank Equivalence (...) This is valid if matrix dimensions align with the expanded coordinate system?*
>
> Yes, this is still valid. In particular, by expanding the number of coordinates, we create $H$ independent blocks to encode assignments. This allows us to one-hot encode the assignments $a_1, .., a_H$, results of $H$ a-queries of the current, each within one of the blocks. In the matrix $W_1$, in each row (corresponding to a potential node $v_{t+1}$, there will be precisely one 1 per assignment block, indicating the unique tuple of answers $a_1, .., a_H$ that lead to the node $v_{t+1}$ from $v_t$.
>
> There is a small typo here, we must have $-H$ before the special coordinate instead of $-(H-1)$. This is because we want the expression $ReLU(b_0 + b_1 + … + b_H - H)$ to be $1$ if and only if $b_0 = b_1 = … = b_H = 1$ if and only if $b_0 + b_1 + … + b_H = H +1$. We will fix this in the revised version.
>
> Re: *the paper does not validate if $t\text{-Comp}$’s structure fully aligns with PC’s assumptions (e.g., cyclic dependencies).*
>
> Let us clarify that for this lower bound, we consider phi’s that map numbers from the first half to the second half, and numbers from the second half to the first half. Such mappings are decomposable into two independent function $g:{1, .., n/2} \rightarrow {n/2+1, … n}$ and $h:{n/2+1, … n}\rightarrow {1, .., n/2}$, which fully aligns the assumptions of the pointer chasing problem. Functions that are not decomposable in this way are not needed.
>
> *Q1:*
>
> The exact relation between hard and soft attention constitutes a major open problem in the area. Recently, there has been some progress with regard to simulating hard attention with softmax, see e.g. Yang et al 2024 [Simulating Hard Attention Using Soft Attention] . Then there are examples of tasks, like PARITY, which can be done with softmax but not with hard attention. At the same time, in experiments softmax transformers struggle to learn PARITY. So lower bounds for hard attention do seem to predict well what a softmax transformer can do in practice, even if these lower bounds do not always generalize in theory.
>
> *Q2:*
>
> Great question! By Yao’s principle, this is equivalent to asking whether there is a randomized transformer that, on any input, solves the task with probability of error at most 1%. We do not immediately see how to extend our lower bound to work against randomized transformers, but we hope that some argument exists. Indeed, this question is relevant from a practical point of view, as real language models generate tokens by sampling.
>
> *Q3:*
>
> We hope that some ``recursive’’ version of EH rank can capture multi-layer hard attention decoders. For instance, we can think of level-2 decision trees as low-rank decision trees that in their leaves, instead of just 0,1, can compute a low-rank decision tree. For instance, the palindrome function can be computed as a disjunction of conjunctions, which is a level-2 decision tree of rank 1, as disjunctions and conjunctions are normal rank-1 functions.  We find it plausible that some variation of this notion will capture 2,3,4…-layer decoders, but of course, this is a future-work direction.
>
> *Q4:*
>
> The matrix W_2 is set to be the identity matrix, as stated on Page 6 line 285. As for W_1, to increase the readability of the proof, we have defined it as a matrix of a linear transformation, defined in (6-8). We believe it is easy to deduce the explicit description of W_1 from the description of the corresponding linear transformation. Namely (W_1)_{ij} is equal to the value of the i-th coordinate of the image of our linear transformation on the vector that has 1 in the j-th coordinate and 0s elsewhere.
>
> *Q5*
>
> We actually cite this paper, but we have to fix the citation, thank you for pointing us out to this issue. By the results of Li et al., to solve P-complete problems, transformers need polynomially many iterations of CoT. In turn, EH rank allows to precisely characterize what functions are computable in any given fixed number of iterations (1, 2, 3, and so on). Results are thus formally incomparable and complement each other in different regimes of the number of CoT steps.

---

> > ### Comment · Reviewer_MAzv · 2025-04-05
> >
> > I thank the authors for taking time and responding to the questions. All the questions are clearly explained. I would greatly appreciate if the authors can respond to limitations as well.

---

> > > ### Author Response · Authors · 2025-04-08
> > >
> > > Thank you, with regard to the limitations, not addressed in our previous response:
> > >
> > > Limitations 1.
> > >
> > > Although our result holds for just a single layer, we believe that it could inspire similar results for multiple layers. For instance, some generalization of the notion of rank could potentially come into play here.
> > >
> > > Limitation 2.
> > > Our result demonstrates that with learnable positional encoding, there always exists a choice of parameters of positional encoding that computes a given function in the number of steps, equal to its rank. We do not immediately see how to obtain our result with sinusoidal positional encoding, this is a great question for future work.
> > >
> > > Limitation 4
> > > Let us also point that tasks, considered in this paper, have some practical motivation, in particular, function composition.  Previous papers, like https://arxiv.org/abs/2311.13314, observed that LLMs struggle with prompts that can be viewed as composition of functions (e.g. ‘’When is Frederic Chopin’s father’s birthday?). Peng et al https://arxiv.org/abs/2402.08164 introduced t-Comp in this context exactly as a model of compositional task, which could be used to explain why LLMs struggle with compositional tasks. We will elaborate on this in the revised version.

---

### Official Review · Reviewer_F5ay · 2025-03-18

**Overall Recommendation:** 3

**Summary:**

This paper characterizes the notion of rank for Boolean and non-boolean functions with a one-layer transformer. Specifically, they show that the rank of a function is equivalent to the minimum number of chain-of-though steps required by a single-layer Transformer with
hard attention to compute the function. They also generalize the definition to the H-head attention. For some compositional tasks, they show $k$-fold function composition necessitates exactly $k$ CoT steps.

**Claims And Evidence:**

The theoretical claims are supported by proofs.

**Essential References Not Discussed:**

The paper discussed several related works. But I think it would be good to discuss the relation with these works. For example, what is the relation between the rank and the different circuit complexity? What is the complexity of the t-Comp and k-thOne?

**Experimental Designs Or Analyses:**

There is no experiments in the paper.

**Methods And Evaluation Criteria:**

They evaluate the proposed rank for simple compositional task t-Comp and k-thOne.

**Other Comments Or Suggestions:**

- line 209 $u_{i_h}$ is not defined.
- line 344 should be the product of $f$ and $g$?

**Other Strengths And Weaknesses:**

### Strengths
- I think the connection between the rank of a function and the steps of CoT is quite interesting. It might help us to have a better understanding of the steps of CoT required for different problems.
-  The proposed connection between rank and decoder depth allows us to have a finer analysis of the task complexity at hand.

### Weaknesses
- The specific tasks considered, t-Comp and k-thOne, are very simple. It would be great to consider some more practical tasks, e.g. the arithmetic tasks and Dynamic Programming in [1]. Or maybe give more examples of functions with different ranks so that the reader can have a better sense of the complexity of different tasks.
- The fact that t-Comp and k-thOne need t or k CoT steps seems not very surprising since every step can only compute a single composition/iteration and the multi-head can only add different functions together but not compose functions.
- It would be great to discuss the relations with previous works. What is the relation between the rank and circuit complexity?


[1] Guhao Feng, Bohang Zhang, Yuntian Gu, Haotian Ye, Di He, and Liwei Wang. Towards revealing the mystery behind chain of thought: A theoretical perspective. NeurIPS, 2023.

**Questions For Authors:**

See questions above.

**Relation To Broader Scientific Literature:**

The key contributions of the paper might help improve the understanding of the expressive power of transformers.

**Theoretical Claims:**

I didn't check the proofs. But the theoretical results make sense to me.

---

> ### Author Rebuttal · Authors · 2025-03-30
>
> Thank you for all your valuable comments. Below we respond to some:
>
> 1. *The specific tasks considered, t-Comp and k-thOne, are very simple. It would be great to consider some more practical tasks, e.g. the arithmetic tasks and Dynamic Programming in [1]. Or maybe give more examples of functions with different ranks so that the reader can have a better sense of the complexity of different task*
>
> Thank you for the suggestions. We will study [1] to apply our technique for the tasks considered there. We can also add more simple examples. For instance, [Hahn, 2020] gives examples of functions (Parity, Majority, Dyck) that are not doable with a constant number of hardmax layers (without CoT).  We can show that these functions have rank, linear in the input length, meaning that they require a linear number of CoT steps for 1-layer hardmax decoders.
>
> [1] Guhao Feng, Bohang Zhang, Yuntian Gu, Haotian Ye, Di He, and Liwei Wang. Towards revealing the mystery behind chain of thought: A theoretical perspective. NeurIPS, 2023.
>
> Let us also point out that tasks considered also have practical motivation. Arguably, compositional/relational tasks are a very important part of human linguistic skills. Previous papers, like https://arxiv.org/abs/2311.13314, observed that LLMs struggle with prompts that can be viewed as composition of functions (e.g. ‘’When is Frederic Chopin’s father’s birthday?). Peng et al https://arxiv.org/abs/2402.08164 introduced t-Comp in this context exactly as a model of  compositional task, which could be used to explain why LLMs struggle with compositional tasks. We will elaborate on this in the revised version.
>
> *2. The fact that t-Comp and k-thOne need t or k CoT steps seems not very surprising since every step can only compute a single composition/iteration and the multi-head can only add different functions together but not compose functions.*
>
> We agree with the reviewer that lower bounds on t-Comp and k-thOne are intuitive. However, we think that this is one of the strengths of our paper, that we provide a formal rigorous proof. With the proof, we can now with 100% certainty say CoT cannot do anything better than compute a single composition per iteration, even with a constant number of attention heads.
>
> *3. It would be great to discuss the relations with previous works. What is the relation between the rank and circuit complexity?
>
> Great question. Ehrenfeucht and Hausler have shown that functions with constant rank have polynomial decision-tree size, which also implies they have polynomial circuit-size. By the result of Liu et al., that implies that problems with constant rank are solvable with polynomially many CoT steps. Our results clarify this by showing that only a constant number of steps suffices in this case.
> We will also add a reference to the following recent paper [On (simple) decision tree rank, Dahiya, Mahajan] that has some connections of the rank with other complexity measures.
>
> Regarding some minor comments:
>
> *line 209 $u_{i_h}$ is not defined.*
>
> this is a typo, it has to be x_{i_h}, thanks for noticing!
>
> *line 344 should be the product of $f$ and $g$?*
>
> sorry, the phrase ``Namely, by the product of g : A → B and h: A → C'' should be ''Namely, by the product of f : A → B and g: A → C''

---

> > ### Comment · Reviewer_F5ay · 2025-04-05
> >
> > Thanks for the response. I would like to increase my score to 3. I encourage the authors to include the discussion and more examples in the revised paper.

---

### Official Review · Reviewer_QZRi · 2025-03-21

**Overall Recommendation:** 3

**Summary:**

The authors study the expressivity of single-layer transformers with hard attention by studying the question: How many chain of thought steps are required to compute particular functions that map strings of length n to some finite output set. The Ehrenfeucht-Haussler rank of a Boolean function measures the complexity of Boolean decision trees that compute the function. The authors introduce a generalisation of the Ehrenfeucht-Haussler rank for non-Boolean functions and obtain various results that relate the number of chain of thought steps required to compute a function to its rank. They first establish that the rank of a function is identical to the number of chain of thoughts required to compute the function. They also give a generalisation of this result to transformers with multiple attention heads by defining a corresponding H-head rank of a function. For particular examples, they establish that the ranks of an n-fold composition function and the function computing the position of the n:th 1 of an input is n.

Finally they show that the H-head rank can be at most H times smaller than the 1-head rank, and that H-head ranks of the n-fold composition function and the function computing the position of the n:th 1 of an input is n. Hence, for these function adding the number of heads does not decrease the number of chain of thoughts required.

## update after rebuttal
I maintain my previous assessment.

**Claims And Evidence:**

The paper is well written and presented. The claims made in the paper are supported by clear intuitive explanations and formal proofs.

**Essential References Not Discussed:**

I am not aware of related works that should be cited.

**Experimental Designs Or Analyses:**

NA.

**Methods And Evaluation Criteria:**

NA.

**Other Comments Or Suggestions:**

Please in the future use two-column line numbering.

-l.145, right. Typo: "and out task".
-l.175. It is a bit out of place to define [n] here, since the notation has been used already many times before in the paper.
-l.277. Typo: "the t-the"

**Other Strengths And Weaknesses:**

This is a solid paper. The study of the expressivity of transformers is an important and timely topic, and this paper takes an interesting angle to this by relating to complexity of decisions trees needed to compute corresponding functions. The results are interesting and non-trivial.

**Questions For Authors:**

No questions.

**Relation To Broader Scientific Literature:**

The authors do a good job in setting the scene of the paper and in positioning their results with respect to literature.

**Theoretical Claims:**

The proof ideas seem plausible. I did not identify any technical issues, but I did not check all the technical details.

---

> ### Author Rebuttal · Authors · 2025-03-28
>
> Thank you for the positive review and for noticing some typos!

---

### Decision · Program_Chairs · 2025-05-01

**Decision:**

Accept (poster)

**Comment:**

This is a theoretical work providing theorems and insights on simple transformers. The high-level message is that there is a close connection between the minimal number of chain-of-thought steps a single-layer hard-attention Transformer needs and the function's rank (a well-studied complexity measure, mainly in the context of learning decision trees). Extensions to multi-head attention are also provided.
Overall, the reviewers were positive about this work and uniformly supported the paper's acceptance.